# A Novel Ship Collision Avoidance Awareness Approach for Cooperating Ships Using Multi-Agent Deep Reinforcement Learning

Chen Chen [1], Feng Ma [2,*], Xiaobin Xu [3], Yuwang Chen [4] and Jin Wang [5]

1   School of Computer Science & Engineering, Wuhan Institute of Technology, Wuhan 430205, China; chenchen0120@wit.edu.cn
2   Intelligent Transportation System Center, Wuhan University of Technology, Wuhan 430063, China
3   School of Automation, Hangzhou Dianzi University, Hangzhou 310018, China; xuxiaobin1980@hdu.edu.cn
4   Alliance Manchester Business School, University of Manchester, Manchester M13 9PL, UK; Yu-wang.Chen@manchester.ac.uk
5   School of Engineering, Liverpool John Moores University, Liverpool L3 3AF, UK; J.Wang@ljmu.ac.uk
*   Correspondence: martin7wind@whut.edu.cn; Tel.: +86-027-8658-2280

**Abstract:** Ships are special machineries with large inertias and relatively weak driving forces. Simulating the manual operations of manipulating ships with artificial intelligence (AI) and machine learning techniques becomes more and more common, in which avoiding collisions in crowded waters may be the most challenging task. This research proposes a cooperative collision avoidance approach for multiple ships using a multi-agent deep reinforcement learning (MADRL) algorithm. Specifically, each ship is modeled as an individual agent, controlled by a Deep Q-Network (DQN) method and described by a dedicated ship motion model. Each agent observes the state of itself and other ships as well as the surrounding environment. Then, agents analyze the navigation situation and make motion decisions accordingly. In particular, specific reward function schemas are designed to simulate the degree of cooperation among agents. According to the International Regulations for Preventing Collisions at Sea (COLREGs), three typical scenarios of simulation, which are head-on, overtaking and crossing, are established to validate the proposed approach. With sufficient training of MADRL, the ship agents were capable of avoiding collisions through cooperation in narrow crowded waters. This method provides new insights for bionic modeling of ship operations, which is of important theoretical and practical significance.

**Keywords:** multi-agent deep reinforcement learning (MADRL); Deep Q-Network (DQN); maritime autonomous surface ships (MASS); multi-ship cooperative collision avoidance; reward function

## 1. Introduction

Under the trend of economic globalization, maritime transportation system plays an important role in the international supply chain [1–4]. Especially maritime autonomous surface ships (MASS) are considered as a promising area in the maritime transportation system. In 2018, the Maritime Safety Committee (MSC) of the International Maritime Organization (IMO) defined the objectives, concept, degrees of autonomy, methodology and work plan of maritime autonomous surface ships (MASS) [5]. MASS can offer a perfect solution to the dilemma of the modern shipping industry, where safety is still the primary concern. Intelligent collision avoidance is a key ingredient for MASS, involving hazard identification, collision avoidance and maneuvering decision making. However, in formal systems research, ship collision avoidance methods are usually applicable on the condition that only the "own ship" is intelligent. This means that only the own ship makes decisions, and other ships are regarded as obstacles that always keep their motion status. Nevertheless, achieving collision avoidance is actually the result of cooperative

behaviors by multiple ships. Therefore, it is necessary to simulate the actions of multi-ship cooperative collision avoidance.

In this research, multi-agent deep reinforcement learning (MADRL) was used to address the problem of intelligent collision avoidance and cooperation modeling. In general, reinforcement learning (RL) can be considered as a method of mapping from environment to appropriate behaviors. An agent seeks a promising action by maximizing the corresponding value function, which is similar to the profit and loss consideration or balance of manual works. On this basis, cooperative collision avoidances among multiple ships can be modeled as the profit and loss allocation of decision making among multiple RL agents. Moreover, navigation conventions and personalities of ship operators can be described as different reward functions in terms of collisions, cooperation and competition. After sufficient training, the artificial consciousness of ship collision avoidance is capable of making safe decisions and control, even if there is no cooperation between ship agents at all.

In order to achieve this goal, a novel multi-ship collision avoidance approach based on MADRL was proposed which takes ship maneuverability into consideration in this research. The paper is organized as follows: Section 2 briefly reviews relevant references. Section 3 puts forward a novel MADRL-based approach. This approach is validated in Section 4 through a simulation case study. Section 5 concludes this approach and provides directions for future research.

## 2. Literature Review

### 2.1. Ship Collision Avoidance Methods

In general, artificial ship collision avoidance mainly depends on ship position and motion relationship to determine the collision avoidance opportunity and make collision avoidance decisions using methods such as a ship domain-based approach [6], time to the closest point of approach (TCPA) and distance at closest point of approach (DCPA) [7]. Collision avoidance of an unmanned surface vessel (USV) depends on sensor fusion methods [8–11], which are capable of detecting static and dynamic obstacles.

Autonomous collision avoidance decision making of USV draws on the methods of robot collision avoidance. A* and B-spline [12], Artificial Potential Field (APF) [13], a modified Dijkstra algorithm [14] and an ant colony optimization method [15] were suggested for obstacle detection and avoidance. The evidential reasoning theory was used to evaluate collision risks [16] to make collision avoidance decisions. The anti-collision system of USV was built on a neural-evolutionary fuzzy algorithm [17] and an evolutionary neural network [18].

With the development of RL, Chen et al. [19] proposed an approach based on Q-learning for smart ships without any prior knowledge. Zhao and Roh [20] came up with an obstacle avoidance model based on deep reinforcement learning (DRL). Chen et al. [21] made use of a Deep Q-Network (DQN) to avoid collisions.

Traditional multi-agent collaboration problems are generally addressed by distributed constraint optimization (DCOP) [22]. DCOP refers to a distributed constrained optimization problem that decision variables and mathematical constraints are distributed in different individuals. Li et al. [23] applied this method to multi-ship collision avoidance, predicting ship trajectories based on ship dynamics, giving different candidate rudder angles, evaluating the collision risk by each rudder angle, and then using optimization strategies to find the most effective collision avoidance plan for ships. Lisowski [24] used a multistep matrix game model to address the problem of autonomous robots' collision avoidance. However, in this research, all ships were controlled by a system decision module, and each agent had no independent decision-making intelligence.

Collective motions are widespread in nature, such as the concerted movements of fish, ants, birds, etc. A number of relevant studies applied swarm control to multi-robot, unmanned vehicle formation control, crowd evacuation, etc. There are many models about collective motions, while a leader-follower model is one of the most widely applied. This

method adopts a centralized control structure, where one agent is the leader and the other agents are followers. The leader-follower method is widely applied to design formation control for USVs [25,26]. Wang et al. [27] proposed a distributed DRL algorithm for USV formations, which is capable to arbitrarily increase the number of ships or change formation shapes. The individual intelligence in swarm dynamics is simple. Zhou et al. [28] made use of the DRL for USV formation path planning. However, this kind of formation control is often very different from real multi-ship collision avoidance, since any single agent in this research does not realize independent decision making.

### 2.2. Multi-Agent Deep Reinforcement Learning (MADRL)

With the success of DRL, it has been applied to multi-agent systems, and MADRL has been developed. MADRL is a stochastic game-based Markov decision-making process [29], which can be described as a tuple $(n, S, A_1, \ldots, A_n, T, \gamma, R_1, \ldots, R_n)$, where $n$ is the number of agents, $S$ is a finite set of environment states, $A = A_1 \times \ldots \times A_n$ is the collection of action sets, $A_1, \ldots, A_n$, one for each agent in the environment. $T$ is the state transition probability function, controlled by the current state $S$ and one action from each agent: $T : S \times A_1 \times A_2 \ldots \times A_n \to S'[0, 1]$. $\gamma$ is a discount rate, $0 \leq \gamma \leq 1$. $R$ is the return function, $R_i$ is the reward of agent $i$ in state $S$ after taking joint action in state $S'$.

In the multi-agent case, state transitions are the result of the joint action of all the agents. The policies $M_i : S \times A \to M$, form the joint policy $M$ together. Suppose in the $t$th episode, the environment's current state is $S_t = s$. An agent $i(i = \{1, 2, \cdots, n\})$ selects and performs an action $A_{t,i} = a$ and observes the subsequent state $S_{t+1}$. Accordingly, the reward of agent $i(i = \{1, 2, \cdots, n\})$ is:

$$R_i^M = E[R_{t+1}|S_t = s, A_{t,i} = a, M] \tag{1}$$

The Bellman equation is

$$v_i^M(s) = E_i^M[R_{t+1} + \gamma V_i^M(S_{t+1})|S_t = s] \tag{2}$$

where the value of a state, $v_i^M(s)$, is the total expected discounted reward attained by the optimal policy $M$ starting from state $s$.

$$Q_i^H(s, a) = E_i^M[R_{t+1} + \gamma Q_i^M(S_{t+1}, A_{t+1})|S_t = s, A_t = a] \tag{3}$$

where $Q_i^H(s, a)$ is the total expected discounted reward [19].

According to different rewarding schemes, different games can be created, such as a fully cooperative one, a fully competitive one and a transition between cooperation and competition, which is also called mixed games.

Collaborative agents performed better than an independent agent in experiments [30]. Tampuu et al. [31] extended the DQN algorithm to multi-agent environments in the Pong videogame, with the two agents controlled by independent DQN. By manipulating reward rules, they demonstrated how competitive and collaborative behaviors emerge. MADRL has reached the level of professional players in a first-person multiplayer game and cooperated with other real players [32].

As discussed above, this research used MADRL to realize the cooperative collision avoidance of multiple ships and raise the awareness of human decision makers. Each ship was regarded as an agent which observes the state of itself and the others as well as the surrounding environment, judges the navigation situation and makes decisions accordingly in the multi-ship encounters. In addition, different agent reward function schemas were designed to simulate the states of a cooperation mode, such as a fully competitive one, a fully cooperative one, and a transition between cooperation and competition. Finally, repeated training was carried out in different encounter scenarios to realize the cooperative collision avoidance among multiple ships.

### 2.3. Literature Summary

In accordance with the literature discussed previously, state-of-the-art collision avoidance of ships can be concluded as follows: Many efficient methods have been invented to model collision avoidance, but few of them can describe the cooperative relations among ships directly, which is inconsistent with the reality. RL uses reward exchanging to simulate negotiation among agents, which has been proven to be effective in many other areas. It is a promising way to describe cooperation among ships in collision avoidance.

## 3. A Proposed Approach

### 3.1. Flow Chart

This research proposes a novel approach to model the cooperation among ships using RL and reward exchanging. The flow chart of the discussed algorithm and its algorithmic presentation is showed as show in Figure 1.

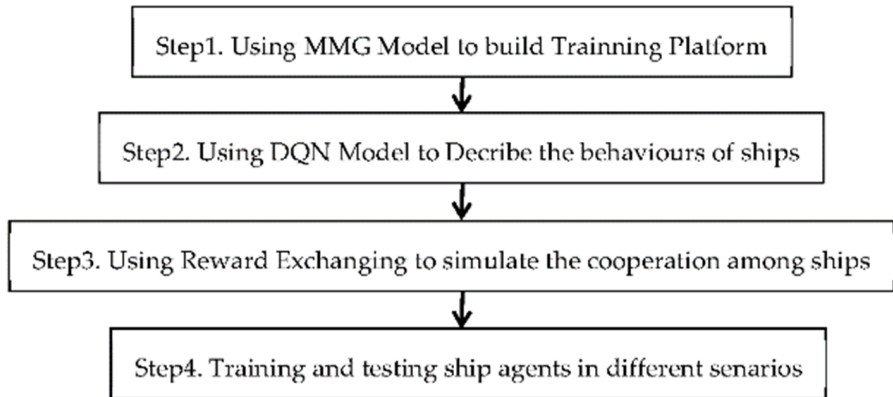

**Figure 1.** Flow chart.

### 3.2. Mathematical Modeling of Ship Motions

Ship maneuvering motions are dynamic models for predicting a ship's trajectory [33] while making the simulated environment consistent with the real world. In similar studies, the MMG model, a mathematical model, is used to express ship maneuvering motions, ref. [21] which take surge, sway, and yaw into considerations. Figure 2 describes the static earth-fixed $o_0 - x_0 y_0 z_0$ and the dynamic body-fixed $o - xyz$ coordinate systems. The origin of $o - xyz$ is located at the middle of the ship $O$. $x$-, $y$ - and $z$-axes are positive to the bow of a ship, the starboard of the ship, and downwards from the water surface $xy$, respectively. Assuming that the ship presented in Figure 2 is maneuvering at surge speed $u$ and sway speed $v$, the ship speed is $V = \sqrt{u^2 + v^2}$. The heading angle is $\psi$. Turning this ship at a rudder angle $\delta$, the yaw rate is denoted as $r = \dot{\psi}$.

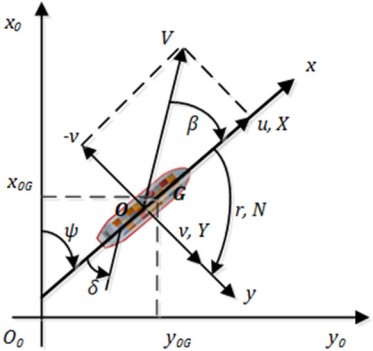

**Figure 2.** Definitions of ship motions.

In this research, the MMG model was used to predict the coming status of a ship when a specific action has been taken. The equations are given as follows [21]:

$$
\begin{cases}
(m + m_x)\dot{u} - (m + m_y)vr - x_G mr^2 = X_H + X_P + X_R \\
(m + m_y)\dot{v} + (m + m_x)ur + x_G m\dot{r} = Y_H + Y_R \\
(I_Z + x_G^2 m + J_Z)\dot{r} + x_G m(\dot{v} + ur) = N_H + N_R
\end{cases}
\tag{4}
$$

In these equations, $H$, $P$ and $R$ mean the status of hull, propeller and rudder, respectively, with force ($X$ and $Y$) and moment ($N$). $m$ denotes the mass of this ship, $m_x$ and $m_y$ denote the extra mass in surge and sway directions due to motions in corresponding directions. $\dot{u}$, $\dot{v}$ and $\dot{r}$ denote acceleration of surge, sway and yaw, respectively, and $I_z$, $J_z$ are the moments of inertia. In particular, $I_Z \approx (0.25 L_{pp})^2 m$, where $L_{pp}$ is the distance between perpendiculars [34]. Generally, the parameters are defined on or around midships.

### 3.3. Markov Decision Process of Multi-Ship Cooperative Collision Avoidance

For multi-ship cooperative collision avoidance, each ship is an agent capable of observing the environment, collecting data and autonomous learning. Its state space is formulated on the current rudder angle, position, speed and heading of each ship, which can be represented by

$$
S = [angle_1, x_1, y_1, v_1, \psi_1, angle_2, x_2, y_2, v_2, \psi_2, \cdots, angle_n, x_n, y_n, v_n, \psi_n]
\tag{5}
$$

where $n$ is the number of the agents, $x$ is the X-coordinate, $y$ is the Y-coordinate and $\psi$ is the heading of the ship. For simplification of the model and computing, the speed of the simulated ship is set to be constant.

In order to simplify the model and make convergence faster, this research defined the action space as [−5, 0, 5], meaning that the rudder angle turns 5° to the left, remains unchanged, or 5° to right, respectively. In fact, other rudder angle values are also applicable. Too many action candidates would make convergence long. Considering the rudder angle of a cargo ship generally does not have many choices, within ±35°. Therefore, the rudder angle after taking an action must also be within [−35°, 35°]. Therefore, the steering angle of a ship is generally between ±35°, and the rudder angle after taking an action must also be within this range.

For a multi-ship system, it is necessary to define the reward value of a single agent first. In fact, too many factors affect the decisions of a helmsman. Hence, this research only chose five typical factors for simplicity from different perspectives, aiming to demonstrate the applicability of the proposed approach.

(1) Approaching a destination. Generally speaking, each ship should reach its destination. If the ship cannot approach the destination, the navigation is considered as failure. This reward was set as

$$
a_{destination} =
\begin{cases}
\lambda_{destination}, & \text{approaching the destination} \\
-\lambda_{destination}, & \text{else}
\end{cases}
\tag{6}
$$

where $\lambda_{destination}$ is a constant greater than 0. When the ship agent is approaching its destination, the reward is set to $\lambda_{destination}$. When the ship cannot approach its destination, the reward is set to $-\lambda_{destination}$. This policy will encourage the ship not to deviate from the navigation destination.

(2) Lane deviation. Lane deviation is an abnormal behavior which easily leads to accidents. Therefore, lane deviation is not encouraged while navigating. Hence, this reward was denoted as

$$
a_{lane} =
\begin{cases}
\lambda_{lanein}, & \text{in lane} \\
-\lambda_{laneout}, & \text{lane deviation}
\end{cases}
\tag{7}
$$

where $\lambda_{lanein}$ denotes the reward when sailing in a waterway normally, and $-\lambda_{laneout}$ denotes the punishment when the ship has sailed out of the route.

(3) Ship domain. Ship domain defines an area which should not be invaded by other ships for safety [10]. In practice, a ship domain is generally seven times the ship's length and three times the width. A helmsman should always avoid entering another ship's domain. This reward was denoted as

$$a_{danger} = \begin{cases} -\lambda_{danger}, & \text{in ship domain} \\ 0, & \text{else} \end{cases} \tag{8}$$

where $-\lambda_{danger}$ denotes the punishment when some other object enters the ship's domain.

(4) Collision. Avoiding collision should always be the priority. Hence, in collisions, whether with ships, bridges or shallow waters, the agent should be punished. This reward was denoted as

$$a_{collision} = \begin{cases} -\lambda_{collision}, & \text{if collision} \\ 0, & \text{else} \end{cases} \tag{9}$$

where $-\lambda_{collision}$ denotes the punishment value. When collision happens in a simulation, the whole training process should be reloaded. To make collision avoidance more important to ships, the parameter $\lambda_{collision}$ should be very large.

(5) Avoidance rules. Ship collision avoidance rules are very complex. This research selected one typical rule for modeling. The ships tends to avoid the coming ship from its right side and to sail through the stern of the other ship. The avoidance of violating this process can be regarded as unreasonable. Other rules or conventions can also be modeled by this method.

$$a_{regulation} = \begin{cases} -\lambda_{regulation}, & \text{breaking the rule} \\ 0, & \text{else} \end{cases} \tag{10}$$

where $-\lambda_{regulation}$ denotes the punishment when a ship is not following the rules.

Taking all the factors into consideration, the total reward can be defined as

$$a = a_{destination} + a_{lane} + a_{danger} + a_{collision} + a_{regulation} \tag{11}$$

Moreover, these factors were set preliminarily, where $\lambda_{destination}$ = 1000, $\lambda_{lanein}$ = 10, $\lambda_{laneout}$ = 100 and $\lambda_{danger}$ = 100, $\lambda_{collision}$ = 500, $\lambda_{regulation}$, = 200. As elaborated previously, many more factors affect the ship in the real world. We chose these factors for simplicity.

### 3.4. Different Cooperative Relationships Between Ship Agents

Compared to a single agent, each agent is affected not only by the environment, but also by other agents in a multi-agent system. Therefore, each agent in a multi-agent system must observe the state and behavior of other agents, and the state transition and reward value of each agent are affected by the joint action of all agents.

Similarly, each ship agent must observe the state and action of other agents, and their own state and behavior will also affect other agents in the multi-ship system. This research assumed that there were two ship agents in the system. When these two ships encounter each other, the two agents will be in different cooperative relationships, making different decisions if their cooperation goals are different.

(1) Fully cooperative:

Each agent not only considers its own navigational safety, but also avoids putting the other one in danger during an encounter. Such two ship agents are fully cooperative. To achieve this goal, both agents are penalized whenever one agent is in danger. In other words, the goal of the two ship agents is to maximize the sum of their cumulative returns.

(2) Fully competitive:

On the contrary, agents only focus on their own safety, even if their decisions will put the other in danger. The goal of the two ships is to maximize each one's own cumulative returns, regardless of the reward value and safety of the other. Such two ship agents are fully competitive.

(3) A mixed game:

When two ships encounter each other, they form the relationship of a transition between cooperation and competition if they are neither fully cooperative nor fully competitive.

Suppose the two ship agents' rewards are $a_1$ and $a_2$, which can be calculated by Equation (11) after performing a certain action. $\rho_1$ and $\rho_2$ are assumed as cooperation coefficients.

The return function of Ship 1 can be defined as

$$R_1 = a_1 + \rho_2 a_2 \tag{12}$$

Accordingly, the return function of Ship 2 can be defined as

$$R_2 = \rho_1 a_1 + a_2 \tag{13}$$

Then, the return function of the system is the sum of the reward functions of the two agents,

$$R = R_1 + R_2 \tag{14}$$

As shown in Table 1, when $\rho_1$ and $\rho_2$ are both equal to 1, the return function can be maximized only when both ships obtain positive returns, and the two agents are fully cooperative. While $\rho_1$ and $\rho_2$ are both equal to 0, each agent only considers maximizing its own reward, and the two agents are fully competitive. While $\rho_1$ and $\rho_2$ are from 0 to 1, the two agents are in a mixed stochastic game.

**Table 1.** Cooperative relationships between multi-ships.

| $\rho_1$ | $\rho_2$ | Cooperative Relationships |
|---|---|---|
| 1 | 1 | Fully cooperative |
| 0 | 0 | Fully competitive |
| [0,1] | [0,1] | Mixed game |

It is appropriate to simulate and learn the decision making of crew members with different personalities in multi-ship encounters in this way. As a result, agents can select optimal actions in different modes.

### 3.5. The Network Structure of a Multi-Ship Cooperative System

As discussed previously, the collision avoidance among ships is quite a multi-dimensional and complex social behavior which should take many factors into considerations. In traditional research, this issue is usually treated as multiple individual subproblems, and the results are generally not satisfactory, since the cooperation among ships is quite difficult in modeling. Neural networks have been proven to be efficient in handling multi-dimensional inputs and cooperation among agents, which can treat the inputs and processing together. Therefore, neural networks were used here for modeling actions, observations and cooperation.

As shown in Figure 3, the multi-agent network was modeled by a multi-layer perceptron. The input of the system was its state space, represented by $[angle_1, x_1, y_1, v_1, \psi_1, angle_2, x_2, y_2, v_2, \psi_2]$. There were 128 nodes in its first layer, which was fully connected. The second layer was also a fully connected layer, with 64 nodes. The output layer consisted of three nodes, corresponding to the three actions of action space.

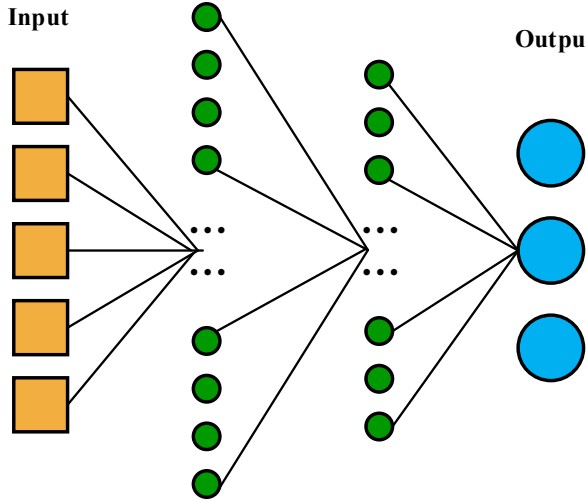

**Figure 3.** The network model of the multi-ship system.

To ensure the stability of convergence, the DQN algorithm was adopted, which stores the current state, action, return value and next state in a replay buffer, sampling through the greedy policy [35], which is a deterministic policy of making the locally optimal choice at each stage with the hope of finding a global optimum. Only when the value function was maximized, the probability was 1 and the other action's probability was set to be 0.

The goal of the system was to make the difference between the target Q network and the Q network as little as possible. More importantly, each ship agent was controlled by the DQN algorithm with the same structure and parameters. The training parameters of its network model are shown in Table 2.

**Table 2.** Hyper parameters of the training algorithm.

| Parameter | Value |
|---|---|
| Learning Rate | 0.0002 |
| Discount Rate | 0.99 |
| Minibatch Size | 128 |
| Replay Memory Size | 20,000 |
| Target Network Update Frequency | 1000 |
| Initial exploration | 1 |

## 4. A Case Study and Validation

### 4.1. Experimental Platform

To verify the effectiveness of the proposed approach, PyCharm [19] was used to establish a simulation environment. As discussed previously, this research only used two agents to reduce calculation and to speed up convergence. Moreover, the two ship agents chose a KVLCC2 tanker [34] as the motion model, which is the standard object of modeling in navigation studies. Simulations were performed with the model-scale ship parameters as presented in Table 3.

**Table 3.** Basic parameters of the KVLCC2 within the MMG model.

| Attributes | Value |
|---|---|
| Length (m) | 7 |
| Width (m) | 1.17 |
| Draught (m) | 0.46 |
| Block coefficient (-) | 0.81 |
| Propeller revolution per second (1/s) | 10.4 |
| Range of rudder angles (deg) | −35~35 |

A scenario editor was designed and developed based on Pygame and Tinker [36]. In this scenario editor, it is possible to set the scenario size, ship size, departure, destination, ship speed, etc. Moreover, the reward function can be set for each ship agent based on the description in Section 3 in this scenario editor.

According to the International Regulations for Preventing Collisions at Sea (COL-REGs), this scenario editor modeled three scenarios, head-on, overtaking and crossing [15].

### 4.2. Training in Different Scenarios

The training was carried out separately with three different scenarios. As discussed previously, cooperative and competitive agents emerged by adjusting the cooperation coefficient of the two ship agents. The video of the trained ship sailing cooperatively in different scenarios can be found online (https://www.youtube.com/watch?v=h7ssNImWECg&list=PLia6EPeX0ULyw6FRlo0MZyYGiC9rlze-C).

### 4.2.1. Head-On

This scenario size was set to 240 × 560 pixels, where the top-left corner was taken as the origin (0, 0). The initial position of Ship 1 was (120, 30), and its destination was (120, 560), while the initial position of Ship 2 was (120, 560), and its destination was (120, 0). The speed of the two ship agents was 1.0 pixels per second, with initial heading angle set to 0. It was found that the two agents were capable of avoiding collision only in the fully cooperative scheme after training. Due to the narrow waterway, two ship agents in the fully competitive and mixed games could not spare enough space for each other. Hence, it was difficult to avoid collision and impossible to sail safely. Based on Figure 4a, it can be inferred that both ship agents turned to the left in the head-on encounter. When one ship agent left the domain of the other, they both turned starboard and returned to the middle of the waterway.

### 4.2.2. Overtaking

The overtaking encounter scenario size was also set to 240 pixels × 560 pixels, where the top-left corner was taken as the origin (0, 0). The initial position of the two ship agents was (120, 480), and their destination was (120, 0). The speed of the ship agent overtaking was 1.5 pixels per second, while that of the ship being overtaken was 0.4 pixels per second.

Similarly, it was found that the two agents were capable of avoiding collision only in the fully cooperative scheme after sufficient training. Based on Figure 4b, it can be inferred that the ship being overtaken turned starboard while the overtaking ship turned to the left in the overtaking situation. When the overtaking process was over, the overtaken ship turned left and returned to the middle of the waterway.

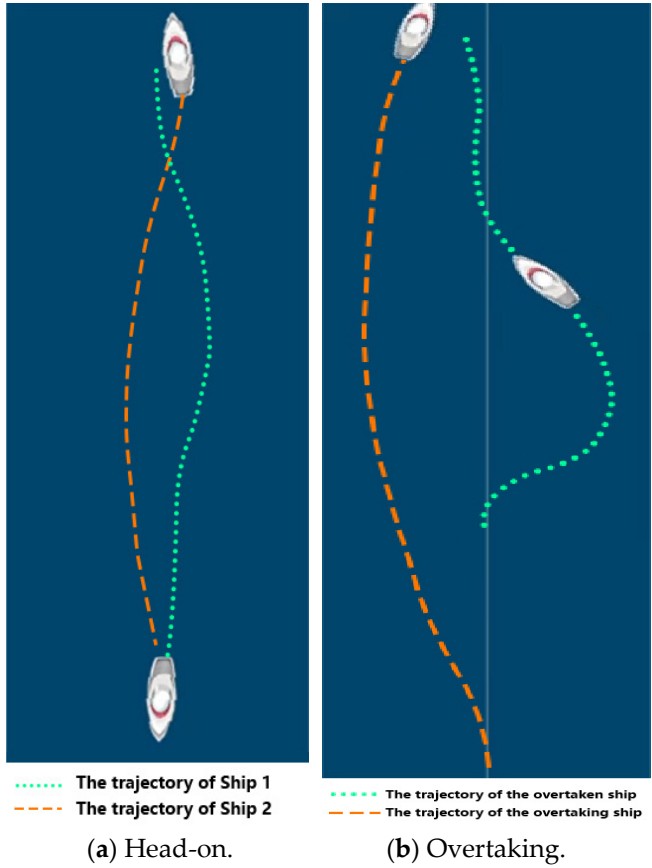

(**a**) Head-on.                    (**b**) Overtaking.

**Figure 4.** The trajectories of ship agents in the collision avoidance process.

### 4.2.3. Crossing

The size of the crossing encounter scenario was set to 480 pixels × 480 pixels, where the top-left corner was taken as the origin (0, 0). The initial position of Ship 1 was (240, 0), and its destination was (240, 480), while the initial position of Ship 2 was (0, 240), and its destination was (480, 240). The speed of the two ship agents was 1.0 pixels per second with the initial heading angle set to 0. In this scenario, the multi-agent system had acquired cooperative collision avoidance intelligence through training in three cooperative schemes.

(1) Fully cooperative:

As discussed above, when both $\rho_1$ and $\rho_2$ were set to 1, the two agents were fully cooperative, and the goal was to achieve the optimal return value of the two agents. From Figure 5a it can be seen that both ships turned starboard and passed through the port side of each other. Furthermore, Ship 1 passed through the stern of Ship 2. It can be concluded that the collision avoidance of the two ships followed "right hand collision avoidance", which met the requirement of COLREGs.

(2) Fully competitive:

Both $\rho_1$ and $\rho_2$ were set to 0, the two agents only took their own safety and efficiencies into consideration. From the experimental results, both agents turned left and passed through the starboard side of the other one, and Ship 1 passed through the bow of Ship 2, as shown in Figure 5b. Although collision avoidance was successful, it did not conform with the navigation rules, and was still very dangerous in practice.

(3) Mixed game:

In this experiment, both $\rho_1$ and $\rho_2$ were set to 0.5. As a result, the two agents played a mixed game. From the experimental results, both agents turned left and passed through the starboard side of the other one, and Ship 1 passed through the bow of Ship 2, as shown in Figure 5c. The collision avoidance process of the two ships also went against collision

convention. However, the "dangerous situation" did not appear, since the two agents took early action.

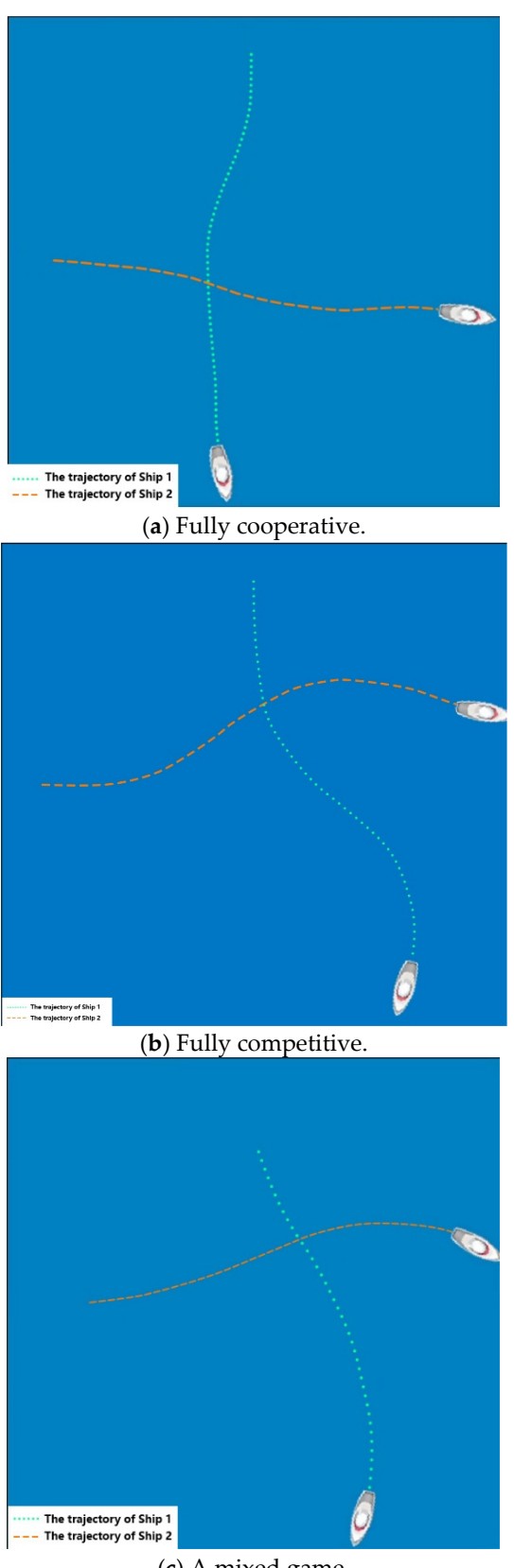

(**a**) Fully cooperative.

(**b**) Fully competitive.

(**c**) A mixed game.

**Figure 5.** The trajectories of ship agents in collision avoidance process of crossing.

## 5. Conclusions

In order to simulate the cooperative collision avoidance awareness between multi-ships, this research analyzed the cooperation mechanism between agents using MADRL and established several cooperative schemas by determining the coefficient in reward functions. In line with the rules of ship collision avoidance, this study modeled different scenarios and verified the proposed method. Differently from the traditional nonlinear optimization-based method, each MADRL agent was featured with an independent operation consciousness and capable of making relatively reasonable decisions even without direct cooperation and intervention of the other agents, which is highly similar to the human consciousness. Overall, this research provided new insights for bionic modeling of ship operations, which is of important theoretical and practical significance.

The limitations of this study can be concluded as follows:

(1) The testing scenarios should be expanded with static obstacles, wind and currents being taken into consideration.
(2) The action spaces of ships should include different engine speeds.

Moreover, it was found that increasing number of agents led to exponential growth of the action space, which made the training computationally expensive in a more complex avoidance experiment. Therefore, it is necessary to develop new methods to reduce computational complexity. On the other hand, it might be more efficient to incorporate human knowledge into the MADRL-based model in order to speed up convergence in finding the optimal route.

**Author Contributions:** Conceptualization, F.M. and Y.W.; methodology, C.C. and F.M.; validation, C.C. and A.M.; writing—original draft preparation, C.C. and F.M.; writing—review and editing, X.X., Y.C. and J.W.; supervision, Y.C. and J.W.; project administration, X.X.; funding acquisition, X.X. All authors have read and agreed to the published version of the manuscript.

**Funding:** This research was supported by Zhejiang Province Key R&D projects (2021C03015), NSFC-Zhejiang Joint Fund for the Integration of Industrialization and Informatization (U1709215), Zhejiang outstanding youth fund (R21F030005), High-tech Ship Research Projects Sponsored by MIIT- Green Intelligent Inland Ship Innovation Programme, National Key R&D Program of China [2018YFB1601503] and Ministry of Industry and Information Technology of the People's Republic of China [2018473].

**Conflicts of Interest:** The authors declare no conflict of interest.

## Abbreviations

| | |
|---|---|
| AI | Artificial intelligence |
| RL | Reinforcement Learning |
| DRL | Deep Reinforcement Learning |
| MADRL | Multi-agent Deep Reinforcement Learning |
| DQN | Deep Q-Network |
| COLREGs | International Regulations for Preventing Collisions at Sea |
| MASS | Maritime Autonomous Surface ships |
| USV | Unmanned Surface Vessel |
| MSC | Maritime Safety Committee |
| IMO | International Maritime Organization |
| TCPA | Time to the Closest Point of Approach |
| DCPA | Distance to the Closest Point of Approach |
| APF | Artificial Potential Field |
| DCOP | Distributed Constraint Optimization |
| MMG | Mathematical Model Group |

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
