# Peer review of "A Novel Ship Collision Avoidance Awareness Approach for Cooperating Ships Using Multi-Agent Deep Reinforcement Learning"

_jmse, doi:10.3390/jmse9101056_

Round 1

Reviewer 1 Report

Article is interesting, the subject is current and has useful value. In order to enhance the article quality, I suggest the following remarks be taken into account:

  1. The literature review should be extended to include the latest publications on intelligent solutions in shipping that refer to the discussed subject, for instance:
    • Borkowski, P.; Pietrzykowski, Z.; Magaj, J. The Algorithm of Determining an Anti-Collision Manoeuvre Trajectory Based on the Interpolation of Ship’s State Vector. Sensors 2021, 21, 5332.
    • Huang, Y.; Chen, L.; Chen, P.; Negenborn, R.R.; van Gelder, P.H.A.J.M. Ship Collision Avoidance Methods: State-of-the-art. Saf. Sci. 2020, 121, 451–473.
    • Lisowski, J. Synthesis of a Path-Planning Algorithm for Autonomous Robots Moving in a Game Environment during Collision Avoidance. Electronics 2021, 10, 675.
  2. Why is assumed the action space as [-5, 0, 5]?
  3. Please consider change the notation the agent reward "r" for a different sign (letter) - earlier in the paper the "r" is reserving for rate of turn.
  4. I suggest that for a better understanding of the paper content and for an easier implementation of the proposed algorithm it would be necessary to rewrite the section 3 by including a flowchart of the algorithm and its algorithmic presentation with all the steps that need to be taken.
  5. Please provide used rewards and punishments values. The authors should add an impact study for different settings of this values.

Reviewer 2 Report

This study proposes a novel ship collision avoidance awareness approach for cooperating ships using multi-agent deep reinforcement learning. I think the paper fits well the scope of the journal and addresses an important subject. However, a number of revisions are required before the paper can be considered for publication. There are some weak points that have to be strengthened. Below please find more specific comments:

*I suggest adding a sentence or two in the abstract to highlight the outcomes of this work and contributions to the state-of-the-art.

*The manuscript contains quite a few abbreviations. I suggest creating a table or an appendix that clearly defines all the abbreviations used.

*In the introduction section starts with a discussion regarding the ship safety. Before talking about the ship safety, the authors should create a more general discussion that highlights the importance of liner shipping for the economic development of numerous countries and the growing demand for effective maritime transportation. This discussion should be supported by the following relevant references:

Akbulaev, N. and Bayramli, G., 2020. Maritime transport and economic growth: Interconnection and influence (an example of the countriesin the Caspian sea coast; Russia, Azerbaijan, Turkmenistan, Kazakhstan and Iran). Marine Policy, 118, p.104005.

Pasha, J., Dulebenets, M.A., Kavoosi, M., Abioye, O.F., Theophilus, O., Wang, H., Kampmann, R. and Guo, W., 2020. Holistic tactical-level planning in liner shipping: an exact optimization approach. Journal of Shipping and Trade, 5(1), pp.1-35.

Bagoulla, C. and Guillotreau, P., 2020. Maritime transport in the French economy and its impact on air pollution: An input-output analysis. Marine Policy, 116, p.103818.

Dulebenets, M.A., 2018. A comprehensive multi-objective optimization model for the vessel scheduling problem in liner shipping. International Journal of Production Economics, 196, pp.293-318.

Dui, H., Zheng, X. and Wu, S., 2021. Resilience analysis of maritime transportation systems based on importance measures. Reliability Engineering & System Safety, p.107461.

This will definitely improve section 1 and make it more solid.

*Please check the alignment of equations throughout the entire manuscript. The equations should align similarly to the alignment of the general text.

*In the literature review section, I suggest creating a subsection 2.3 Literature Summary and Contributions. This subsection should provide a concise summary of the state-of-the-art, the major gaps, and how these gaps are addressed by the present study.

*There are quite a few equations in the manuscript and many of them are not supported by the relevant references. Please include the supporting references for the adopted equations where appropriate to justify their selection.

*Section 3 should include a more detailed discussion and explain why neural networks were used for modeling purposes.

*The conclusions section should expand on limitations of this study and future research needs. I suggest listing the bullet points.

Round 2

Reviewer 1 Report

My suggestions provided in my original review have been incorporated in the manuscript. From my side the work is accepted in this new version.

Reviewer 2 Report

The authors have adequately addressed my original concerns regarding the manuscript. The quality and presentation of the manuscript have been improved. Therefore, I recommend acceptance.